Cryoelectrolysis—electrolytic processes in a frozen physiological saline medium

Lugnani Franco 1
Macchioro Matteo 2
Rubinsky Boris rubinsky@berkeley.edu brubinsky@gmail.com 3
1 Clinica Santa Elena , Malaga , Spain
2 Hippocrates D.O.O , Divaca , Slovenia
3 Department of Bioengineering and Department of Mechanical Engineering, University of California , Berkley , CA , United States
Diao Jiajie
Electronic publication date: 2017 Jan 17
Publication date: 2017
Volume: 5
Electronic Location ID: e2810
Received 2016 Sep 17; Accepted 2016 Nov 21
Copyright: ©2017 Lugnani et al.
Copyright year: 2017
Copyright holder: Lugnani et al.
License: This is an open access article distributed under the terms of the Creative Commons Attribution License, which permits unrestricted use, distribution, reproduction and adaptation in any medium and for any purpose provided that it is properly attributed. For attribution, the original author(s), title, publication source (PeerJ) and either DOI or URL of the article must be cited.
License URL: https://creativecommons.org/licenses/by/4.0/

Keywords: Cryoelectrolysis, Cryosurgery, Electrolysis, Electrolytic ablation, Freezing, Electro-osmosis, Iontophoresis, Cancer ablation, Focal therapy ablation, Minimally invasive surgery

Funding: The authors received no funding for this work.

==============================
Background

Cryoelectrolysis is a new minimally invasive tissue ablation surgical technique that combines the ablation techniques of electrolytic ablation with cryosurgery. The goal of this study is to examine the hypothesis that electrolysis can take place in a frozen aqueous saline solution.

Method

To examine the hypothesis we performed a cryoelectrolytic ablation protocol in which electrolysis and cryosurgery are delivered simultaneously in a tissue simulant made of physiological saline gel with a pH dye. We measured current flow, voltage and extents of freezing and pH dye staining.

Results

Using optical measurements and measurements of currents, we have shown that electrolysis can occur in frozen physiological saline, at high subzero freezing temperatures, above the eutectic temperature of the frozen salt solution. It was observed that electrolysis occurs when the tissue resides at high subzero temperatures during the freezing stage and essentially throughout the entire thawing stage. We also found that during thawing, the frozen lesion temperature raises rapidly to high subfreezing values and remains at those values throughout the thawing stage. Substantial electrolysis occurs during the thawing stage. Another interesting finding is that electro-osmotic flows affect the process of cryoelectrolysis at the anode and cathode, in different ways.

Discussion

The results showing that electrical current flow and electrolysis occur in frozen saline solutions imply a mechanism involving ionic movement in the fluid concentrated saline solution channels between ice crystals, at high subfreezing temperatures. Temperatures higher than the eutectic are required for the brine to be fluid. The particular pattern of temperature and electrical currents during the thawing stage of frozen tissue, can be explained by the large amounts of energy that must be removed at the outer edge of the frozen lesion because of the solid/liquid phase transformation on that interface.

Conclusion

Electrolysis can occur in a frozen domain at high subfreezing temperature, probably above the eutectic. It appears that the most effective period for delivering electrolytic currents in cryoelectrolysis is during the high subzero temperatures stage while freezing and immediately after cooling has stopped, throughout the thawing stage.

Introduction

Tissue ablation with minimally invasive and non-invasive methods has emerged as an important branch of surgery. Various physical and chemical phenomena are used to ablate tissue, each with their advantages and disadvantages, and particular applications. For example, thermal ablation with nanoparticles (Kennedy et al., 2011), thermal ablation with radiofrequency electromagnetic waves (Gazelle et al., 2000), thermal ablation with freezing, cryosurgery (Rubinsky, 2000), chemical ablation that employs the products of electrolysis (Nilsson et al., 2000) and non-thermal irreversible permeabilization of the cell membrane, non-thermal irreversible electroporation, (Rubinsky, 2010). Recently, our group has become involved in studying combinations of these ablation techniques. The combinations examined include: electrolysis and electroporation; cryosurgery and electroporation; and cryosurgery and electrolysis (Lugnani et al., 2015; Rubinsky et al., 2015; Stehling et al., 2016). This paper pertains to the latter, the combination of cryosurgery and electrolysis, termed cryoelectrolysis (Lugnani et al., 2015), which is a largely unexplored process. Cryoelectrolysis, is marked by its potential to utilize the advantages of both cryosurgery and electrolytic ablation while overcoming their disadvantages. First, a brief review on the principles and attributes of cryosurgery and electrolytic ablation when used separately, followed by the principles of cryoelectrolysis and a description of the hypothesis examined in this work.

Cryosurgery

Cryosurgery is the ablation of undesirable tissues by freezing (Rubinsky, 2000). The procedure employs a cryogenic fluid internally cooled cryosurgical probe, inserted in the undesirable tissue. The freezing propagates from the cryoprobe surface outward to freeze and, hopefully, thereby ablate the entire undesirable tissue. An important finding in cryosurgery is that the extent of freezing can be monitored in real time, by essentially every medical imaging techniques (Gilbert et al., 1984; Onik et al., 1984; Rubinsky et al., 1993). This facilitates real time control over the extent of freezing. However, it was also found that cells can survive freezing at high subzero freezing temperatures. Therefore, cells can survive on the outer rim of the frozen lesion or around blood vessels, within the frozen lesion. Thus, the extent of freezing seen on medical imaging does not correspond to the extent of cell death. Currently, to increase the probability that all the cells in the frozen lesion are ablated, surgeons employ two to three cycles of freezing and thawing, which makes the procedure excessively long. Also, attempts are made to enhance cell death throughout the frozen lesion, by chemical means (Baust et al., 1997; Koushafar et al., 1997; Koushafar et al., 1997; Clarke et al., 2001; Mir & Rubinsky, 2002). A disadvantage of the chemical methods is the need to inject chemicals in the treated volume; a procedure that suffers from lack of control and precision.

Electrolytic ablation

Electrolytic ablation, also known as Electro-Chemical Therapy (EChT), is a tissue ablation technique that employs products of electrolysis for cell ablation (Nilsson et al., 2000). In EChT a direct electric current is delivered to the treatment field through electrodes that are inserted in the treated tissue. New chemical species are generated at the interface of the electrodes and tissue as a result of the electric potential driven transfer between the electrode electrons and ions or atoms in the tissue. The various chemical species produced near the electrodes diffuse away from the electrodes, into tissue, in a process driven by differences in electrochemical potential. Tissue ablation by electrolysis is caused by two factors: the cytotoxic environment developing due to local changes in pH, as well as the presence of some of the new chemical species formed during electrolysis. Electrolytic ablation requires very low direct currents (tens to hundreds of mA) and very low voltages (single to low tens of Volts) (Nilsson et al., 2000). This is advantageous, because it makes the devices used for this technology extremely simple and safe. However, the procedure is long, from tens of minutes to hours. The length is related to the slow diffusion of electrochemically produced species in tissue and the need for high concentrations of electrolytic products to cause cell death. A clinical study on tissue ablation with electrolysis states that— “Currently, a limitation of the technique is that it is time consuming” (Fosh et al., 2002; Fosh et al., 2003).

Cryoelectrolysis

The idea for tissue ablation by cryoelectrolysis, i.e., a combination of cryosurgery and electrolytic ablation, emerged from fundamental studies on the process of freezing in physiological saline solutions (Rubinsky, 1983; Rubinsky & Ikeda, 1985; Rubinsky et al., 1987; Rubinsky & Pegg, 1988; Rubinsky et al., 1990; Ishiguro & Rubinsky, 1994). Figure 1 is a compendium of data from a number of our earlier studies and is brought here in a modified form, to facilitate a better understanding of the concept. Figures 1A–1D illustrate a series of events that occur on the solid–liquid interface during the solidification process in physiological saline. These events are driven by a thermodynamic condition known as constitutional supercooling (Rubinsky, 1983). Constitutional supercooling predicts that even in a one dimensional solidification process, the solid/liquid change of phase interface is thermodynamically unstable and cannot remain planar. The sequence of Figs. 1A–1C show how the interface becomes perturbed during the freezing process. Finger like ice crystals form and develop as dendritic structures. Ice has a very tight crystallographic structure and cannot contain any solutes. Therefore, the solutes previously contained in the volume now occupied by ice gather in the liquid between the ice crystal fingers. Figure 1D shows the ultimate outcome of the freezing process in saline. High concentration brine solutions reside between finger like ice crystals. The concentration of the brine increases towards lower temperatures, until it reaches the eutectic at about −21.1°. Figures 1E–1G show results from experiments in which we froze saline solutions with red blood cells. Figure 1E is from the higher temperature tip of the finger like ice crystal structures. Figure 1F is for a lower temperature and Fig. 1G is a further lower temperature. The white arrows point to the brine channels. It is evident that as the temperature decreases the volume of the channels decrease and concentration of brine increases. Figure 1H is a low temperature scanning electron micrograph of frozen liver. Here, ice forms inside the blood vessels (BV) and sinusoids (s) and the concentrated brine (light areas) surrounds the ice crystals and is in contact with the cells. The white arrow points to the concentrated brine and cells.

Figure 1 Compendium of schematic and experimental results to serve as an explanation for the fundamental concepts of cryoelectrolysis.

This figure is a compendium of unpublished data from one of the authors BR.

Cryoelectrolysis combines cryosurgery with electrolysis to overcome the limitations of cryosurgery and electrolysis used separately. The idea for the concept of cryoelectrolytic ablation was inspired by the findings described above, namely, that freezing of tissue increases the concentration of solutes around cells, by removing the water from the solution in the form of ice (Rubinsky & Pegg, 1988). Freezing also causes cell membrane lipid phase transition, disrupts the cell membrane lipid bilayer and causes it to become permeabilized (Mir & Rubinsky, 2002). From the data in Fig. 1, it occurred to us that freezing of tissue in the presence of products of electrolysis will increase the concentration of the products of electrolysis around the cell. Furthermore, freezing induced cell membrane permeabilization will expose the interior of cells to the products of electrolysis and enhance cell death. The permeabilization of the cell membrane by freezing should decrease the concentration of the electrolytic products needed to cause cell death. Because the production of the electrolytic products is a time dependent reaction, decreasing the amount of electrolytic compounds needed for cell ablation, should shorten the time of an electrolytic induced mechanism of cell ablation. This is the basic principle of the cryoelectrolytic ablation concept proposed in Lugnani et al. (2015). In that concept, the targeted tissue is first treated with electrolysis to generate products of electrolysis in the targeted volume; after which the targeted tissue is frozen to increase the local concentration and the exposure of the cell interior to the products of electrolysis in the frozen lesion. Theoretically the cryoelectrolysis combination should require lower concentrations of products of electrolysis i.e., shorter period of electrolysis and only one freeze thaw cycle. This should yield a shorter procedure than conventional electrolytic ablation or multiple freeze-thaw cycles of cryosurgery and, increase cell ablation in the frozen lesion by the dual mechanisms of freezing and electrolysis in the frozen lesion. The ability to image the extent of the frozen region, combines the advantages of real time image monitoring of cryosurgery with enhanced cell ablation by the combination freezing and electrolysis, in the frozen region.

Our first study on cryoelectrolysis was designed to examine the hypothesis that the combination of electrolysis and freezing, delivered as described above, is more effective at cell ablation than either electrolysis or freezing alone. The first study employed a protocol in which electrolysis was delivered first, followed by freezing. Experiments on animal tissue have confirmed our hypothesis and have shown that cryoelectrolysis is more effective at cell ablation than either cryosurgery or electrolytic ablation, alone (Lugnani et al., 2015).

While a protocol that employed first electrolysis and then freezing is faster than conventional electrolysis or the use of several freeze thaw cycles in conventional cryosurgery, the study in this paper was designed to explore an idea that may lead to a protocol that may be even faster. We think that the time of the procedure would be shorter, if, electrolysis and freezing, which are both diffusion limited processes, could be done simultaneously. The idea for this new concept was inspired by the same known, fundamental observation, described in regards to Fig. 1; that freezing of tissue increases the concentration of solutes around cells, by removing the water from the solution in the form of ice (Rubinsky & Pegg, 1988). These high concentration of solutes form brine channels within the frozen tissue (Rubinsky et al., 1987; Rubinsky et al., 1990; Ishiguro & Rubinsky, 1994). The hypothesis that we have set to examine in this study is that the channels of high concentration brine in a frozen saline medium could serve as electrical conduits for the process of electrolysis. Therefore, while ice is not electrically conductive, electrolysis could be done through the high concentration brine channels in the frozen region, simultaneously with freezing and thawing.

Materials and Methods

The goal of this study is to examine the hypothesis that electrolysis can occur in frozen saline. In our study we employed a physiological saline gel to simulate tissue and used a modified commercial cryosurgery probe to deliver both cold and to serve as the electrolysis probe. The extent of freezing was monitored visually through change in opacity during freezing and the extent of the electrolysis was monitored also visually using a pH dye. Voltage and electrical current was measured throughout the experiments, to ascertain if and how electrical current flows through the frozen medium.

Materials

A physiological saline based agar was used to simulate tissue. One liter of water was mixed with 9 grams NaCl and 7 grams of agarose (UltraPure Agarose, Invitrogen). The solution was stirred and heated for 10 min and then removed from heat. Two pH indicator dyes were added after five minutes of cooling. For analysis of electrolysis near the anode, methyl red (Sigma-Aldrich®, St. Louis, MO, USA), 1 mL per 100 mL agar solution, was used. For analysis of electrolysis near the cathode we used Phenolphthalein Solution 0.5 wt. % in Ethanol (Sigma-Aldrich) at a concentration of 5 ml per liter agar (or 1 ml per 100 ml agar solution) solution. The agar was cast in a 20 cm diameter cylindrical glass vessel whose radial walls were coated with a 200 µm thick copper foil. The height of the gel cast is 4 cm.

Experimental devices and set-up

The two panels in Fig. 2, show photographs of the experimental setup. For the cryoelectrolysis experiment we used a Endocare® R2.4 cryoprobe with a diameter of 2.4 mm connected to an Endocare® single port control console device regulating flow duration and monitoring feed-back temperatures (Endocare Inc., Austin, TX, USA). The probe is supplied by a pressurized Argon gas container through the control console, at a constant pressure of 3,000 psi. The cooling of the Endocare® stainless steel cryoprobe is through a Joule-Thomson internal valve. The cooling process is typical to all Endocare® cryoprobes of this type. The probe temperature reaches −180 °C, at a rate of cooling governed in part by the thermal environment in which the probe is inserted. A 30 µm foil of gold was wrapped several times around the cryoprobe, to minimize the participation of the electrode metal in the process of electrolysis. The metal body of the probe was connected to a DC power supply (Agilent E3631A; Agilent, Santa Clara CA, USA), to also serve as an electrolysis electrode. In a typical experiment the cryoelectrolysis probe was inserted vertical into the center of the gel. The electrical circuit consists of the power supply, the cryoelectrolysis probe electrode in the center of the gel, the gel and the copper electrode around the gel vessel. The gel was infused with methyl red when the cryoelectrolysis probe served as the anode and with phenolphthalein when the probe served as a cathode. A 1 mm T type thermocouple (Endocare®) was inserted to the vicinity of the cryoprobe at a distance of less than 5 mm from the outer surface of the probe, as shown in Fig. 2. The temperature was recorded continuously, throughout the experiment. It should be emphasized that this is not the temperature at the probe, but rather in the gel at a distance from the probe. A camera was focused on the experimental setup to continuously record the position of the change of phase interface, the position of the pH front, the voltage, current and time.

Figure 2 (A) Photograph of experimental system: a, electrode on container surface; b, cryoelectrolysis probe; c, DC power supply; d, thermocouple, e, camera; f, cryosurgery probe pressure monitor; (B) close-up of the gel and electrodes.

Experimental protocol

In this study, we performed first a number of experiments with electrolysis only, without freezing, to determine the currents and time of application needed to obtain measureable data for the extent of electrolysis in our experimental set-up. From these experiments we chose values of 400 mA, 200 mA and 50 mA. The range of electrical currents tested are typical to clinical electrolytic ablation procedures (Nilsson et al., 2000; Lugnani et al., 2015). Similarly, preliminary experiments were performed with freezing only to evaluate the time of freezing needed to obtained measurable frozen lesions in our experimental configuration. From these results we chose ten minutes of freezing and fifteen minutes for  thawing.

The following experimental procedure was employed in all the experiments. The experimental protocol was designed to examine all aspects of the hypothesis of this study; electrolysis before freezing, electrolysis during freezing and electrolysis during thawing. The electrical circuit, comprised of the cryoelectrolysis probe, the gel and the copper vessel walls, was connected to the power supply first. It remained connected throughout the experiment, during freezing and thawing. The first minute was electrolysis alone. The flow of cryogen began one minute after the circuit was connected to the power supply, initiating the freezing. Constant pressure of 3,000 psi was used to generate the Argon gas flow in a manner typical to clinical cryosurgical treatment with the cryosurgery probe we used. The flow of cryogen was delivered for ten minutes, during which the gel froze. This is the stage in which freezing and electrolysis were delivered simultaneously. After ten minutes, the flow of the cryogen was stopped and the frozen lesion was left to thaw, in situ. The electrical circuit remained connected to the power supply for additional 15 min after the flow of the cryogen was stopped. This represents the stage in which thawing and electrolysis occurs  simultaneously.

We performed three repeats of each experiment with 400 mA, 200 mA and 50 mA currents for both the central electrode anode and the central electrode cathode for a total of 18 cryoelectrolysis experiments with the protocol described above. The voltage was allowed to change to provide the desired current. However, the saturation voltage of the power supply used in this study is 25 V and the system cannot provide a higher voltage. Therefore, when changes in resistance demanded a voltage higher than 25 V, the current dropped and eventually stopped.

Results and Discussion

The primary goal of this study is to examine the hypothesis that electrolysis can occur in a frozen aqueous saline solution. We will bring here results that support the hypothesis.

Figure 3, presents a compilation of photographs that illustrate several important observations, typical to all the experiments performed in this study. Figures 3A and 3B, are images of the progression of the pH front during a preliminary study in which there was only electrolysis, without freezing. The goal of these two panels is to illustrate the appearance of a typical process of electrolysis in a pH stained gel. The cryoelectrolysis probe served as the anode and delivered 400 mA. Figure 3A, shows the radially symmetric pH front around the anode. The panels show a cylindrical pH stained region around the probe. This is the region in which the products of electrolysis reside. The interface between the stained and unstained regions is referred in this paper as the, pH front. The process of electrolysis was continued for several minutes and Fig. 3B shows the extent of electrolysis at a later time. Obviously the pH front has advanced, while remaining radially symmetric. The white line points to an observation of importance to cryoelectrolysis. Diffusion and iontophoresis driven electro-osmosis, are the physical mechanisms that cause the propagation of the pH front from the electrode outward. The electro-osmotic flow is an important aspect of electrolytic ablation in tissue (Lugnani et al., 2015; Phillips et al., 2015a; Phillips et al., 2015b; Rubinsky et al., 2015; Rubinsky et al., 2016). The flow is from the anode to the cathode. The white line points to a dark gap that has formed between the electrode and the gel. (Inserts in Fig. 3 are magnified views of the region near the electrode) The gap was caused by the electro-osmotic driven flow of solution, away from the anode, towards the cathode. The later panels in this figure will illustrate the significance of this electro-osmotic flow to cryoelectrolysis.

Figure 3 Illustration of typical cryoelectrolysis process.

Photographs of the pH front and freezing front in different experiments: (A) electrolysis only, 400 mA current, (B) electrolysis only, 400 mA current at a later time from (A), (C) cryoelectrolysis with cryoelectrolysis probe as the anode, 400 mA, (D) cryoelectrolysis with cryoelectrolysis probe as the anode, 400 mA pH front and ice front at a later time from (C), (E) cryoelectrolysis with cryoelectrolysis probe as the cathode, 50 mA, (F) cryoelectrolysis with cryoelectrolysis probe as the cathode, 50 mA, pH front and ice front at a later time from (E). Top photo earlier time. Bottom photo later time Black arrow—pH front, black dashed arrow—ice front, white line—interesting feature near the cryoelectrolysis probe. Photographs (A)–(B), (C)–(D) and (E)–(F), are to the same scale.

Figures 3C and 3D, are images of the progression of the pH stained region and of the frozen region during a typical cryoelectrolytic protocol of the type described in the materials and methods section. The cryoelectrolysis probe served as anode and delivered 400 mA. Figure 3C shows the appearance of the frozen lesion at the end of the freezing stage of the protocol. The dashed arrow point to the edge of the frozen lesion. Figure 3D is a photograph from the same experiment taken several minutes after the cooling was stopped, while the power supply continued to deliver current to the electrical circuit. Two interesting observations emerge. While the extent of the frozen lesion in Fig. 3D has not changed from that in Fig. 3C; the pH stained region has expanded beyond the frozen lesion. This demonstrates that the process of electrolysis can occur through ice, during the thawing stage. Similar observations were made with all the currents tested and in all the repeats. This is an important observation, which will be discussed later in the context of Figs. 4 and 5. The white arrow shows that the electro-osmotic flow generated gap formed between the electrode and the gel during conventional electrolysis, also occurs during cryoelectrolysis. This further strengthens the evidence that electrolysis occurs through a frozen region.

Figure 4 Progression of a pH front and a ice front during a typical cryoelectrolysis protocol.

Results shown as a function of time after the start of the experiment (in minutes); (A) 1 min, (B) 2 min, (C) 3.5 min, (D) 11 min, (E) 12.5 min, (F) 16 min, (G) 18.5 min, (H) 21, (I) 26 min. All the figures are at the same scale (cm scale shown). The margin of the pH front is marked with a dark arrow and of the ice front with a dotted dark arrow. A feature of interest near the cryoelectrolysis probe marked with a white arrow.

Figure 5 Data from an experiment in which the cryoelectrolysis probe served as the anode and the preset current was 200 mA.

(A) the diameter of the ice front (red line) and of the pH front (blue line); (B) current; (C) voltage; (D) overall resistance; (E) temperature, as a function of time in minutes.

Figures 3E and 3F, are images of the progression of the pH front (the pH stained area) and of the ice front (the frozen lesion) during a typical cryoelectrolytic protocol of the type described in the materials and methods section when the cryoelectrolysis probe served as the cathode and delivered 50 mA. Obviously, the appearance of the treated areas in Figs. 3E and 3F is completely different from that in Figs. 3C and 3D. Figure 3E is from an earlier stage of the cryoelectrolysis protocol, during which, both electrical current and cryogen cooling, were delivered by the cryoelectrolysis probe, simultaneously. It is important to observe that both, a pH stained region and a frozen lesion have formed and they propagate away from the probe. However, in the case of a cathode centered electrode, the propagation is in an asymmetric way. The lack of symmetry is evident in comparison with Fig. 3C. The difference is caused by the direction of the electro-osmotic flow, which in this case, is towards the cryoelectrolysis cathode probe. This generates a high flow rate of solution, at the cryoelectrolysis cathode probe—gel interface. We have observed a flow of water gushing out at the interface between the cryoelectrolysis probe and the gel, regardless of the current magnitude used and in all the cryoelectrolysis cathode probe study repeats. The water also contains a mixture of gas (hydrogen from the reduction reaction near the cathode). Evidence of the process can be seen from the red dots spread over the right hand side of the gel (dotted arrow in Figs. 1E and 1G). The red dots are caused by the splashed droplets of high pH fluid. The electro-osmotic pressure has caused various random and detrimental effects, when the cryoelectrolysis probe is the cathode. For higher currents, of 200 mA and 400 mA, the electro-osmotic pressure driven flow has caused fractures and cracks in the gel. For the lower currents of 50 mA it produced the lack of symmetry seen in Figs. 3E and 3F. The electro-osmotic pressure caused events, occur at random and the cracks formation is not predictable.

Figure 3E was taken during the last stage of the experiment; a stage in the typical cryoelectrolysis protocol in which the cooling was stopped and only electrolysis occurs through the frozen region that is thawing. This is at a similar stage in the protocol to that in which the Fig. 3D photograph was taken. Here, we observe that the pH front has propagated irregularly both within and beyond the frozen lesion. The propagation of the pH front occurred while the frozen lesion still exists. This demonstrates that the process of electrolysis can occur through a frozen domain when the cryoelectrolysis probe is either anode or cathode. The lack of symmetry in the appearance of the pH front in Fig. 3F can be, probably, attributed to cracks that form in the gel because of the electro-osmotic pressure. These cracks favor certain directions of propagation of the electrolytic products flow. The magnified insert of the region near the cryoelectrolysis cathode probe provides further evidence on the effect of the electro-osmotic flow. The dark gap between the cryoelectrolysis anode probe and the gel in Figs. 3C and 3D does not form when the cryoelectrolysis probe is the cathode. In fact, the white arrows point to a bulging volume of ice formed in the vicinity of the cryoelectrolysis probe. The insert also shows a crack in the gel, filed with ice. While qualitatively similar results were observed in all the repeats of the cathode centered experiments, the quantitative appearance was different from repeat to repeat because of the random appearance of the electro-osmotic flow generated cracks.

In summary, this part of the study reveals two important physical phenomena related to cryoelectrolysis: (a) electrolysis can occur through a frozen milieu at both, the anode and the cathode, (b) electro-osmotic flows play an important part in the physical events that occur during cryoelectrolysis. Because of electro-osmotic flows the outcome of the procedure, is different between a cryoelectrolysis cathode probe and a cryoelectrolysis anode probe. The results tentatively suggest that it may be beneficial to use for cryoelectrolysis only the anode and employ a surface electrode (similar to that used in radiofrequency ablation) as the cathode.

Figures 4 and 5 are typical to all the anode center experiments of this study. They were chosen to illustrate the events that are relevant to the hypothesis and which occur during a typical processes of cryoelectrolysis. We focus here on the anode center experiments because for this configuration, the results in the different repeats and with the different currents were similar, unlike for the cathode centered experiments. The cathode center experiments were different from experiment to experiment because of the random formation of electro-osmotic flow induced cracks. We will illustrate the observations with results in which the cryoelectrolysis probe was the anode and the current was set to, 200 mA.

Figure 4 is a sequence of images showing the pH front and the ice front at different instances in time during the cryoelectrolysis protocol. Figure 4A shows the appearance of the pH stained region, one minute after the start of the experiment, just prior to the start of the cooling process. Figure 4B shows the appearance of the ice front and of the pH front one minute after the start of freezing and two minutes after the start of the experiment. It is evident from comparison with Fig. 4A that during this one minute of freezing, the ice front and the pH front have both advanced. This is an important observation as it demonstrates that electrolysis occurs during freezing. However, Figs. 4C and 4D show that after one minute of freezing, the pH front stops advancing (no electrolysis) while the ice front propagates further. This shows that there are conditions in which electrolysis does not occur in a frozen solution. Figures 4D to 4I show that after the coolant has stopped flowing through the cryoprobe, the extent of the frozen lesion remains unchanged for a long period of time. However, the extent of the pH dye stained region increases in time and eventually extends beyond the frozen lesion.

Figure 5 is from the same experiment as Fig. 4. It displays, the data measured during that experiment. The panels show, Figs. 5A to 5E: the diameters of the pH stained region and of the frozen lesion, the measured current, the measured voltage, the calculated resistance and the temperature of the thermocouple, as a function of time during the cryoelectrolysis process examined in this study. The first minute of the protocol is electrolysis, without freezing. Figure 5 shows that during this first minute the current is constant at 200 mA, the temperature is constant at 15 °C, the voltage is about 8 V, resistance is constant and the extent of the pH dye stained region increases in time. All these are evidence of a process of electrolysis. Cooling the probe, began one minute after the start of the experiment. As soon as cooling began, the temperature measured by the thermocouple began to drop. (It should be emphasized that the thermocouple is at a distance from the probe, and does not measure the temperature of the probe, which is lower than the thermocouple measurement.) The other curves in Fig. 5 show that the diameter of the freezing zone increases in time, throughout the ten minutes of cooling. During the first minute of cooling (freezing) there is current and the extent of the pH dye stained region increases. The resistance increases, the voltage increases to the maximum that the power supply can deliver (25 V) and the current decreases to zero after about one minute of freezing. The resistance becomes, in fact, infinite after one minute of freezing. Nevertheless, it is important to notice that during the first minute of freezing there is electrolysis and current flows through the frozen lesion. Cooling continues for ten minutes, during which the thermocouple measured temperature drops further, the frozen zone expands and no current flows through the frozen lesion. After ten minutes of cooling, the flow of the cryogen is stopped, while the power supply for electrolysis remains on. The thermocouple reading shows that the temperature in the frozen region begins to increase as soon as the cooling has stopped. However, an interesting phenomenon occurs. The temperature remains at a high subzero value, below the freezing temperature for the remainder of the experiment, i.e., the temperature around the probe (electrode) is below freezing. Visual observation displayed on the top panel in Fig. 5 and in Figs. 4D to 4I show that the extent of the frozen lesion does not change to the end of the experiment. Within a minute after the cooling has stopped and the temperature began to increase, the current increases, the voltage drops and the resistance drops. The pH dye stained region begins to increase and eventually becomes larger than the frozen lesion. Taken together, this presents evidence that electrolysis occurs in the frozen lesion after freezing has stop.

The results displayed in Figs. 4 and 5 are typical to all the anode center experiments. They demonstrate that electrolysis occurs in a frozen saline solution at high subzero temperatures. The results are consistent with the hypothesis and are explained in the formulation of the hypothesis in the introduction and in Fig. 1. The mechanism responsible for electrolysis in a high subzero frozen media is associated with the process of freezing in solutions and tissues, as described in the introduction. Ice has a tight crystallographic structure and cannot contain any solutes. Constitutional supercooling dictates that during freezing of a solution, finger like ice crystals form and the salt is rejected along the ice crystals (Rubinsky, 1983). High concentration salt solutions form along the ice crystals. This phenomenon occurs during freezing of any aqueous medium, in solutions (Ishiguro & Rubinsky, 1994), gels (Preciado et al., 2003) and tissues (Rubinsky & Pegg, 1988). While the electrical conductivity of ice is essentially zero, electrical currents can flow through these high concentration brine channels until the temperature reaches the eutectic −21.1 °C. Figures 1E, 1F and 1G show that as the temperature decreases, the channels become narrower, until eutectic is reached. Eutectic is a solid phase and ionic movement ceases. This explain the observed increase in resistance during the first minute of freezing and the decrease in resistance after cooling has stopped and the temperature of the frozen tissue began to increase. This result is important in designing cryoelectrolysis protocols, because it shows that electrolysis can occur in a frozen domain, only at high subzero temperatures; most likely below the eutectic. Therefore, in cryoelectrolytic ablation, it should be beneficial to reside longer at high subfreezing temperatures during the freezing stage. This is in marked contrast to current cryosurgery freezing protocols in which freezing is done rapidly to low subzero freezing temperatures.

The phenomena observed after cooling has stopped are particularly interesting and of value to designing a cryoelectrolysis ablation protocol. Figure 5 shows that the temperature measured by the thermocouple begins to raise as soon as the cooling stops. However, the measured temperature remains close to, albeit lower, than the phase transformation temperature for most of the remainder of the cryoelectrolysis protocol. This is a phenomenon we have observed and studied in the past (Rubinsky & Cravalho, 1979; Hong & Rubinsky, 1995). To better understand the phenomenon, we bring here Fig. 6. It is a qualitative depiction of results from mathematical analysis of thawing of frozen cylinders in Rubinsky & Cravalho (1979); Hong & Rubinsky (1995). The figure shows that when a frozen domain begins to thaw from the exterior, as is also the case in the cryoelectrolysis protocol, the temperature of the frozen region raises rapidly towards the change of phase temperature. However, the melting, which propagates from the exterior of the frozen domain towards the interior is very slow, relative to the raise of the temperature in the frozen domain. Therefore, the frozen domain, stays at high subfreezing temperatures throughout the process of melting.

Figure 6 Qualitative depiction of the temperature distribution at various times during the thawing of a frozen cylinder of pure water, at an initial temperature of −40 °C, when the outer surface of the cylinder is 10 °C the time of the curves, increases in the direction of the arrow.

The location of the interface between the frozen tissue and the unfrozen tissue domain corresponds to the 0°C isotherm. he domain at a temperature lower than 0°C is frozen. The figure is a qualitative depiction of the results in Rubinsky & Cravalho (1979).

An explanation for this phenomenon was provided first in (Rubinsky & Cravalho, 1979). The phenomenon is related to the fact that the change in enthalpy during phase transition of ice into water is very large relative to the change in enthalpy due to change in the temperature of the ice. Briefly, during melting, heat is extracted from the frozen domain, through the change of phase interface, by the environment surrounding the interface. The temperature of the change of phase interface is fixed by equilibrium thermodynamics of a two phase system at constant pressure. For physiological saline it is −0.56 °C. As long as there is an ice and water mixture in a domain, the temperature of that domain cannot exceed the thermodynamic phase transition temperature of the solution. The phase transformation process (melting) occurs only on the change of phase interface, which propagates very slowly, because the large change in enthalpy involved. Since the enthalpy associated with changes of temperature in the frozen domain are very small relative to the change in enthalpy by phase transformation, the temperature of the frozen region becomes elevated and reaches the phase transition temperature fast, throughout the frozen region; while the region is still frozen (Rubinsky & Cravalho, 1979). Consequently, while the extent of the frozen regions remains essentially unchanged at the end of cooling (Figs. 4E to 4I) the temperature of the frozen region raises to become close and below the change of phase temperature, for a long period of time; Fig. 5E. The temperature measurements in Fig. 5 validate this explanation. The increase in the temperature of the frozen region has several effects. Figure 5 shows that there is a gradual increase in current and a decrease in resistance, soon after cooling stops. Consequently, there is a process of electrolysis, and the pH front expands beyond the margin of the frozen region, while the region is still frozen (Figs. 4E to 4I).

Figure 5 shows that indeed current begins to flow through the high subzero temperature region of frozen gel, soon after cooling stops. Unavoidable, flow of ionic current is associated with electrolysis and this is why the pH front advances while the tissue is still frozen, albeit at high subzero temperatures. The flow of current through the brine channels most likely elevated the local temperature of these channels and may cause local melting and expansion or collapse of the brine channels. It is possible that this phenomenon is responsible for the jumps in voltage measured occasionally (see Fig. 5). We have seen various sudden jumps in voltage, during the period after the cooling has stopped, in all the experiments.

The physiological effects of electrolysis during the thawing process remain to be examined with living tissue. However, we anticipate that the electrolysis during the thawing process will be effective at tissue ablation. While the phenomenon of concentrating the products of electrolysis by freezing, does not occur anymore, the cell membrane is still permeabilized by cold and provides access to the products of electrolysis. Furthermore, the thawing stage during cryosurgery is unavoidable long. Delivering current during that stage may have a dual effect. It may shorten the length of thawing because of the Joule heating effect and enhance cell death by the products of electrolysis. This is why delivering electrolytic currents during the thawing stage of cryoelectrolysis may be desirable.

Conclusion

The primary goal of this study was to examine the hypothesis that electrolysis can occur in frozen aqueous saline. The combined effect of freezing and electrolysis was studied in a tissue simulant made of a physiological solution of agar with pH dyes. The most important finding of this study is that electrolysis can occur in a frozen aqueous saline and the hypothesis is proven. To the best of our knowledge, this is the first time that electrolysis through ice was observed and reported. This finding is valuable for designing cryoelectrolysis protocols. It demonstrates that the processes of freezing and of electrolysis can be done simultaneously. It appears that the most effective period for delivering electrolytic currents is during the high subzero temperatures while freezing and immediately after cooling has stopped, throughout the thawing stage.

Supplemental Information

Supplemental Information 1 This file includes the graphs showing the diameters of the electrolysis and of frozen lesion, the current, the voltage, the resistance and the temperature recorded during the procedure, as a function of time

The second page shows the raw data from which we built the graphs. The entire experiment lasted 26 minutes.

Click here for additional data file.

Supplemental Information 2 Photographs taken during the experiments showing the progression of the electrolysis and ice fronts during the 26 minutes of procedures

In particular: 160720-015-Exp_Cryoelettro shows details of the experiment, with camera, suppliers, probe and saline agar solution; 160720-016-Exp_Cryoelettro is a closer view of the setup. Photos from 160720-030-Exp_Cryoelettro to 160720-055-Exp_Cryoelettro shows the progression mentioned before from minute 0:00 to minute 12:30; each photo is taken every 30 s.

Click here for additional data file.

Supplemental Information 3 Photographs taken during the experiments showing the progression of the electrolysis and ice fronts during the 26 minutes of procedures

Photos from 160720-056-Exp_Cryoelettro to 160720-082-Exp_Cryoelettro shows the progression mentioned before from minute 13:00 to minute 26:00; each photo is taken every 30 s.

Click here for additional data file.

We are grateful to Dr. Liel Rubinsky who did the first work on cryoelectrolysis and whose work planted the seeds for this paper and to Mr. Paul Mikus for useful advice.

Additional Information and Declarations

Competing Interests

Author Contributions

Patent Disclosures

Data Availability

Two of the co-authors, Franco Lugnani and Boris Rubinsky are part of a group of co-inventors that have submitted a patent disclosure on the technology of cryoelectrolysis. They may gain financial benefits from this technology if the patent is allowed. The patent is under “Methods, systems and apparatuses for tissue ablation using electrolysis and cryosurgical techniques/Application type: provisional/16th August 2016”. Matteo Macchioro is an employee of Hippocrates D.O.O, Divaca, Slovenia. Boris Rubinsky is an Academic Editor for PeerJ. Dr. Franco Lugnani is a consultant to Clinica Santa Elena.

Franco Lugnani performed the experiments, analyzed the data, contributed reagents/materials/analysis tools, reviewed drafts of the paper.

Matteo Macchioro performed the experiments, analyzed the data, contributed reagents/materials/analysis tools, prepared figures and/or tables, reviewed drafts of the paper.

Boris Rubinsky conceived and designed the experiments, analyzed the data, wrote the paper, prepared figures and/or tables, reviewed drafts of the paper.

The following patent dependencies were disclosed by the authors:

Methods, systems and apparatuses for tissue ablation using electrolysis and cryosurgical techniques/Application type: provisional/16th August 2016.

The following information was supplied regarding data availability:

The raw data has been supplied as a Supplementary File.

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
