# Peer review of "Cryoelectrolysis—electrolytic processes in a frozen physiological saline medium"

_PeerJ, doi:10.7717/peerj.2810_

## Round 0.1 · original submission · Minor Revisions

· Academic Editor

Minor Revisions

As you can see. three reviewers have commented on your manuscript. Please revise your manuscript according to their comments.

Reviewer 1 ·

Basic reporting

1. This reviewer would like to see a more powerful description of producing and development course of cryoelectrolysis. The generalizations and advantages appear to need more instructions.

2. Regarding as this system, this reviewer would advise authors to find the practical differences between cryoelectrolysis and freezing/electrolysis, especially the practical application, profound physical meaning, conceivable clinical differences.

Experimental design

1. One issue that concerns this reviewer is the option of current and voltage, how to determine the reasonable range?

Validity of the findings

No Comments.

Additional comments

The manuscript deals with a topic that concerns the understanding of the physical and electrochemical processes which occur during cryoelectrolysis, in particular, how solid/liquid phase transformation and electrolysis occur when delivered simultaneously. In my opinion, some revisions may help to improve clarity and understanding of the text.

Reviewer 2 ·

Basic reporting

Minimally invasive and non-invasive surgery is an important branch of surgery. Cryoelectrolysis ablation has emerged as a very useful method which combine the advantages of Cryosurgery and electrolysis ablation. The authors studied the physical and electrochemical in this process using simulate systems. Such study is very essential for our understanding the basic principle for cryoelectolytsis ablation and useful for developing optimal cryoelectrolysis medical treatment protocols.

Experimental design

Experimental Design is very smart and reasonable related to authors' research purpose.

Validity of the findings

The result is solid and very interesting.

Additional comments

Introduction

#1 Line 36, the sentence “While recent studies on animal tissue have demonstrated the advantages of combined cryoelectrolytic ablation over cryosurgery ablation and electrolytic ablation, each used separately, (Lugnani, Zanconati et al. 2015), the fundamental physics of the cryoelectrolytic process was not studied before .This is the first study on the physical and electrochemical processes that occur during cryoelectrolysis.”.

This make me confused. The Lugnani’s work is a combination, but the authors use the word “each used separately”. The second thing is which study is the first study? Whose study?


#2 The last paragraph of Introduction. Who is the first one raises the concept “cryoelectrolytic ablation”? please give the proof. It seems like Lugnai’s work is the first one.

#3 Line 86 “This is the basic principle of cryoelectrolytic ablation”. Please give the citation or is just your hypothesis.

#4 The logical order of Introduction should be raising the knowledge gap then introduce your work related to this gap. So please re-organize the part of Introduction and clearly raise the problem and show the importance of your work.



Materials and Methods

#1 Comment. Please give the title of each paragraph.

#2 Comment. Insert the Figure 1 in the “Results and discussion”.



Results and discussion

#1 Comment. It is very hard to read this part. Please separate paragraphs into groups, each with a title.


#2 Comment. Result is not figure legend, please reorganize your sentence using the style shown in Journal “Peer J”.

#3Comment. If you do not have the evidence, please be cautious of the words using in the dissection, the author’s hypothesis seems like results.

Reviewer 3 ·

Basic reporting

No Comments

Experimental design

Reasonable experimental design;
Sufficient description of their protocals.

Validity of the findings

useful findings

Additional comments

The authors have developped Cryoelectrolysis and observed new physical and electrochemical phenomena of relevance to tissue ablation. These findings might be helpful for cryoelectrolytic ablation surgery protocols. In general, I would suggest publish this paper with the following improvement.

Experiment statistics:
Line 145 to 149: Following 30 experiments to identify the parameters... This paper reports the results from the last series of experiments.
I can not comment too much on the technical side. This seems something a bit subjective. I agree it is impossible to show all experiment results and many incorrect conditions that finally led to the optimal conditions. However, as a systematic experiment, I would suggest some kind of controls and statistics of all the results rather than some final curves. Also it would be better to show some kind of the ranges of conditions in which it worked and in which it did not. Otherwise, people could say anything they want if they are not going to show the true data, i.e. 30 experiments, 300 experiments, 3000 experiments or some words like 'data not shown' or similar. I would suggest scientific papers always avoid such claims, but showing all possible data along within the paper.

Theoritical analysis:
It it nicer if the authors could expplain why the phenomena they observed are reasonable in physical and chemical principle. As the authors claimed their findings will become 'the scientific basis', the authors should at least provide some fundamental discussion on these phenomena rather then simply showing what they observed. There are too many phenomena to describe in daily experiments. The only phenomena that finally become real 'scientific basis' is the fundamental analysis that lead to generalized principles, not simply showing data. The author should also discuss or propse possible approaches how other pepole could potentailly benefit from these findings. It sounds a bit vague just claiming 'may become' something or similar words.

---

## Round 0.2 · accepted · Accept

· Academic Editor

Accept

Your revised manuscript reaches the level of publication.